# Stability and molecular pathways to the formation of spin defects in silicon carbide

Elizabeth M. Y. Lee [1], Alvin Yu [2,3], Juan J. de Pablo [1,4✉] & Giulia Galli [1,2,4✉]

Spin defects in wide-bandgap semiconductors provide a promising platform to create qubits for quantum technologies. Their synthesis, however, presents considerable challenges, and the mechanisms responsible for their generation or annihilation are poorly understood. Here, we elucidate spin defect formation processes in a binary crystal for a key qubit candidate—the divacancy complex (VV) in silicon carbide (SiC). Using atomistic models, enhanced sampling simulations, and density functional theory calculations, we find that VV formation is a thermally activated process that competes with the conversion of silicon ($V_{Si}$) to carbon monovacancies ($V_C$), and that VV reorientation can occur without dissociation. We also find that increasing the concentration of $V_{Si}$ relative to $V_C$ favors the formation of divacancies. Moreover, we identify pathways to create spin defects consisting of antisite-double vacancy complexes and determine their electronic properties. The detailed view of the mechanisms that underpin the formation and dynamics of spin defects presented here may facilitate the realization of qubits in an industrially relevant material.

---

[1] Pritzker School of Molecular Engineering, The University of Chicago, Chicago, IL 60637, USA. [2] Department of Chemistry, The University of Chicago, Chicago, IL 60637, USA. [3] Institute for Biophysical Dynamics and James Franck Institute, The University of Chicago, Chicago, IL 60637, USA. [4] Argonne National Laboratory, 9700 Cass Avenue, Lemont, IL 60439, USA. ✉email: depablo@uchicago.edu; gagalli@uchicago.edu

Solid-state spin defects are an emerging platform with applications in quantum information science, sensing, and metrology[1]. The negatively charged nitrogen-vacancy centers (NV⁻) in diamond and neutral divacancy defects (VV) in silicon carbide (SiC) represent promising examples of defect complexes for use as spin qubits[2–4]. Divacancies in SiC—a widely used material in the semiconductor industry—are particularly attractive, as they offer all-optical spin initialization and readout capabilities[5,6], nuclear spin control[7], and a near-infrared high-fidelity spin-photon interface[8], in addition to long coherence times[9].

The optical and electronic properties of spin defects in SiC, including the silicon vacancy ($V_{Si}$)[10,11], NV centers[12,13], and carbon antisite-vacancy complexes ($C_{Si}V_C$)[14], have been characterized using a variety of techniques. These include density-functional theory (DFT) calculations, electron paramagnetic resonance spectroscopy (EPR), deep-level transient spectroscopy (DLTS), and photoluminescence spectroscopy (PL)[15–21]. Little is known, however, of how to control the selective formation and spatial localization of defect complexes. Addressing this challenge is critical for their integration with optical and electric devices and nanostructures[18].

Divacancies and other spin defects in SiC are generally generated by ion implantation or electron irradiation, followed by thermal annealing[22–26]. The vacancy-to-VV conversion efficiency, however, is low—a few percent or less[22,27]. VV localization is difficult to control, as samples need to be thermally annealed at high temperatures (above ~700 ºC) to generate the defect mobility necessary to create divacancies[8]. Furthermore, the spatial placement of VV appears to depend on the initial distribution of several intrinsic and extrinsic defect species (e.g., vacancies, interstitials, substitutional defects, and dopants) and the Fermi level of the system[27].

The dynamics of vacancy-defect complexes in semiconductors, particularly SiC, present considerable challenges for theory and computation as well. In particular, the activation energies for vacancy migration in SiC can be on the order of several electronvolts[19], thereby limiting the applicability of first-principles molecular dynamics (FPMD) simulations. As a result, prior DFT calculations in SiC have focused mainly on computing defect-formation energies and migration barriers for monovacancies at $T = 0$ K[28–33]. Recent kinetic Monte Carlo simulations considered the dynamics of vacancies in SiC[34], but they did so by invoking a priori mechanisms for defect mobilization.

Here, we study how spin defects hosted in vacancy-defect complexes are formed in cubic, 3C-SiC, by coupling FPMD simulations with a neural-network-based enhanced sampling technique[35,36] to efficiently probe phase space, and we compute the free energies and stabilities of several defects. In some cases, our FPMD results are augmented by classical MD simulations conducted with larger system sizes and over longer timescales. Additionally, we predict previously unidentified spin defects consisting of antisites and double vacancies. We focused on the 3C polytype, given the increasing number of low-cost synthesis strategies developed for high-purity 3C-SiC, e.g., heteroepitaxial growth on silicon substrates[37,38]. Furthermore, 3C-SiC contains a single configuration for each vacancy defect ($V_{Si}$, $V_C$, and VV), and it is thus simpler than other polytypes (e.g., 4H-SiC) that can host multiple configurations. We find that the formation of divacancies is initiated by the migration and association of monovacancies, which occur in the same temperature regime as the crystallographic reorientation of VV. Our results reveal that a divacancy is a thermodynamically stable state, while $V_{Si}$ represents a kinetically trapped state that readily transforms into an intermediate carbon antisite-vacancy ($C_{Si}V_C$) defect. Our simulations also show that divacancy formation can be maximized by

choosing initial conditions corresponding to a large fraction of $V_{Si}$ in the sample to mobilize monovacancies prior to the destabilization of $V_{Si}$.

## Results

**Divacancy dynamics at high temperatures.** The simulation of defect migration in covalently bonded materials, where activation energies may be as large as several electron volts, requires the collection of statistics for ~10–100 ns. Therefore, capturing the migration of vacancies in SiC using straightforward FPMD is at present prohibitively demanding from a computational standpoint. Hence, before conducting FPMD simulations coupled to enhanced sampling techniques, we explored the dynamics of vacancy defects in SiC using an empirical force field[39], based on a widely used Tersoff-type bond-order formalism[40,41]. The force field was selected for its ability to yield temperature-dependent bulk densities and a decomposition temperature in agreement with experiments[42,43] (see Supplementary Fig. 1). We found that classical MD using the empirical force field and FPMD simulations yields qualitatively similar energetics for vacancy migration processes (see Supplementary Note 1 for a detailed comparison between the two methods).

To probe the dynamics of VV and its formation from single-vacancy defects (see Fig. 1), which are only present in low concentrations, we carried out 10-ns-long classical MD simulations of a 4096-atom supercell of 3C-SiC containing a pair of $V_{Si}$ and $V_C$ (see "Methods"). Analyses of the MD trajectories reveal one or more of the following phenomena at temperatures between 1000 K and 1500 K: (1) monovacancy migration (Supplementary Movie 1), (2) VV formation from the pairing of two monovacancies (Fig. 2a–d, Supplementary Movie 3), (3) VV orientational changes (Fig. 2e–h, Supplementary Movie 4), and (4) conversion of $V_{Si}$ into a carbon vacancy–antisite complex, $C_{Si}V_C$ (Fig. 2i, j, Supplementary Movie 2). Below 1000 K, we do not observe any vacancy diffusion, even after continuing our simulations for 100 ns; above 1500 K, divacancies dissociate into $V_{Si}$ and $V_C$, and $C_{Si}V_C$ also dissociates into $C_{Si}$ and $V_C$ (Fig. 2j–l). These results suggest that the formation of a VV is a temperature-activated process. VV generation occurs in the same temperature regime as the single-vacancy diffusion and, surprisingly, by the destabilization of silicon vacancies via the conversion of $V_{Si}$ into $C_{Si}V_C$.

Our results between ~1000 K and 1500 K are consistent with recent measurements performed on irradiated samples of 4H-SiC[44]. These annealing experiments reported that with increasing

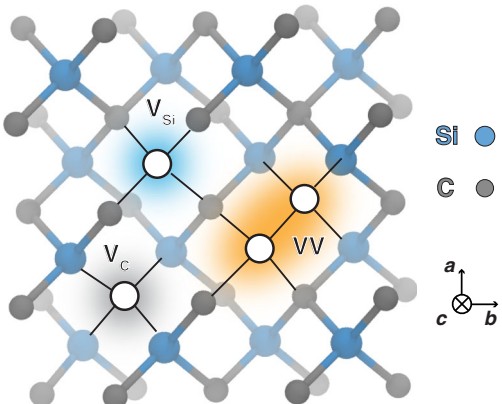

**Fig. 1 Ball-and-stick representation of defects in cubic SiC (3C-SiC).** A divacancy-defect complex (VV) consists of a carbon vacancy ($V_C$) paired with a silicon vacancy ($V_{Si}$). Carbon and silicon atoms are shown as gray and blue spheres, respectively, whereas vacancies are depicted as white solid circles.

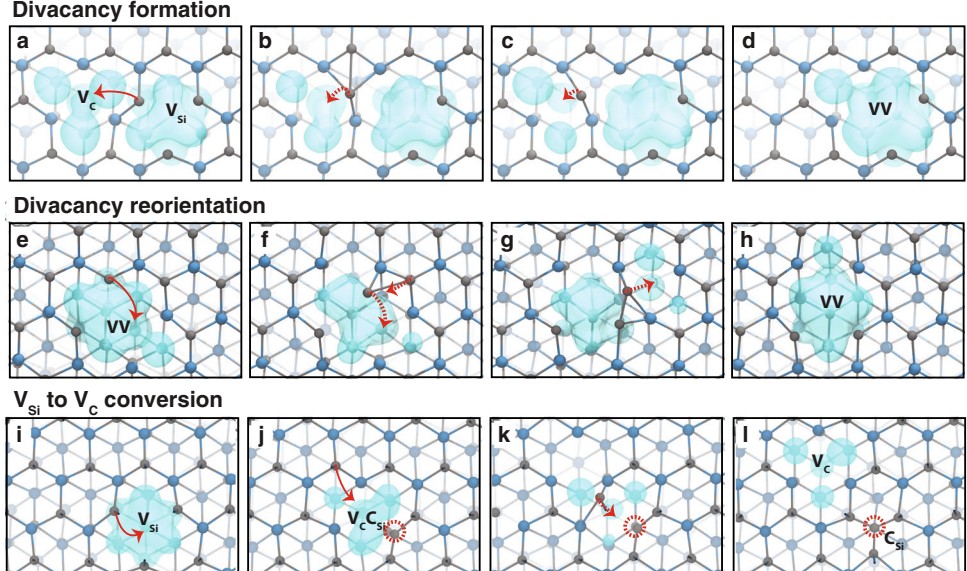

**Fig. 2 Dynamics of vacancy formation and migration from classical molecular dynamics (MD) simulations.** A series of snapshots from classical MD simulations at temperatures above 1000 K show the mechanisms for **a–d**, the divacancy formation, **e–h**, divacancy reorientation, and **i–l**, conversion from a $V_{Si}$ to a $V_C$ before the $V_{Si}$ encounters an existing $V_C$. Three-dimensional vacancy volumes colored in teal blue show the location, size, and shape of void volumes at the defect site. **a** A carbon atom in-between the $V_C$ and $V_{Si}$ sites displaces the $V_C$ and forms a VV, following the path marked by a red arrow. **b**, **c** During intermediate steps, the mobile carbon atom interacts with neighboring atoms as the local coordination changes from five- and threefold. **d** The divacancy consists of a carbon vacancy adjacent to a silicon vacancy. **e** The orientation of VV changes as a carbon atom adjacent to $V_{Si}$ migrates to the $V_C$ site. **f**, **g** C–C bonds are formed and broken during intermediate steps. **h** The $V_{Si}$–$V_C$ axis in the final VV configuration is rotated compared to that in the initial configuration, within the laboratory frame. **i**, **j** $V_{Si}$ converts into $C_{Si}V_C$ as a nearest-neighbor carbon atom displaces it. Red circle indicates the $C_{Si}$ site. **k**, **l** $C_{Si}V_C$ dissociates into $V_C$ and $C_{Si}$ as $V_C$ exchanges position with a nearest carbon atom.

temperature above ~600 K, there is a strong correlation between *decreasing* deep-level transient-spectroscopy (DLTS) intensities of $V_{Si}$ and $C_{Si}V_C$ and *increasing* DLTS and PL intensities of $V_C$ and VV. The experiments, therefore, are also consistent with the formation of VV and the formation of $V_C$ from $V_{Si}$ and $C_{Si}V_C$.

**Mechanism of VV formation and reorientation processes.** Snapshots from our MD simulations above 1000 K are shown in Fig. 2. We provide a 3D representation of the free volume at defect sites (see Methods) to investigate changes to the local structure proximal to each defect. As shown in Fig. 2a, b, the size and shape of the free volume surrounding $V_C$ and $V_{Si}$ are different. In particular, $V_C$ sites have tetrahedrally shaped voids, while $V_{Si}$ sites have void volumes consisting of two overlapping tetrahedra centered at the $V_{Si}$ site. The void volume of the $V_{Si}$ site is larger than that of the $V_C$ site, indicating that silicon and carbon atoms surrounding $V_{Si}$ are more likely to be mobile than the atoms surrounding $V_C$. When an atom in-between $V_C$ and $V_{Si}$ fills in a vacancy site, the two monovacancies join together to form a VV.

Our MD simulations show that the thermal annealing of SiC samples may also lead to a reorientation of the VV without dissociating the original VV, i.e., it leads to a change in the direction of the $V_{Si}$–$V_C$ axis within the laboratory frame. A recent photoluminescence confocal microscopy study of NV⁻ in diamond reported a crystallographic reorientation process, where the NV-defect symmetry axis in a thermally annealed sample was rotated from its starting configuration (before annealing) in the original reference frame[45]. Our results suggest that thermal annealing at temperatures between ~1000 K and 1500 K may be a route toward aligning the orientation of VVs, which would be desirable for sensing applications[46,47].

**Free-energy landscape for divacancy formation and inter-conversions.** To quantify the energetics of defect-interconversion processes and identify possible transition pathways, we computed the free energy or the potential of mean force (PMF) for VV formation at 1500 K and compared the PMF with free-energy profiles of other defect transformation processes that occur at the same temperature. In order to do so, we turned to DFT calculations, and we computed the PMFs by combining a neural-network-based enhanced sampling method with FPMD[36], and we used the Cartesian coordinates of the mobile carbon or silicon atom as order parameters. We performed simulations at a Fermi level known to favor the formation of neutral divacancies (see Methods). As expected, the computed 2D-PMFs for defect transformations in 3C-SiC reveal multiple minima and maxima that depend on the specific location of the defects in the crystal structure (Fig. 3a–c).

In the free-energy profile shown in Fig. 3a and Supplementary Fig. 6a, we identify a transition state (T) separating the free-energy basins of VV and that of the two unassociated vacancies ($V_{Si}$ + $V_C$). We find that the free energy of the VV is ~1.5 eV lower than that of the ($V_{Si}$ + $V_C$), thus confirming that divacancies are stable species at 1500 K, consistent with high-temperature annealing experiments of as-grown 4H-SiC samples[25,48]. Additionally, the free-energy difference between the VV and the transition state is comparable to that of the monovacancy migration barrier (~3.0 eV for the $V_{Si}$ diffusion and ~3.9 eV for $V_C$ diffusion, see Supplementary Fig. 5) as both processes involve a next-nearest-neighbor atom substituting a vacancy site. The predicted $V_C$ migration barrier obtained from FPMD (~3.9 eV) agrees with the barrier measured by DTLS on annealed SiC samples (~3.7–4.2 eV)[19].

Regarding the VV reorientation process, our simulations indicate that the diffusion free-energy barriers of C atoms atoms (Fig. 3d, e) are lower than that of the Si atom (Supplementary

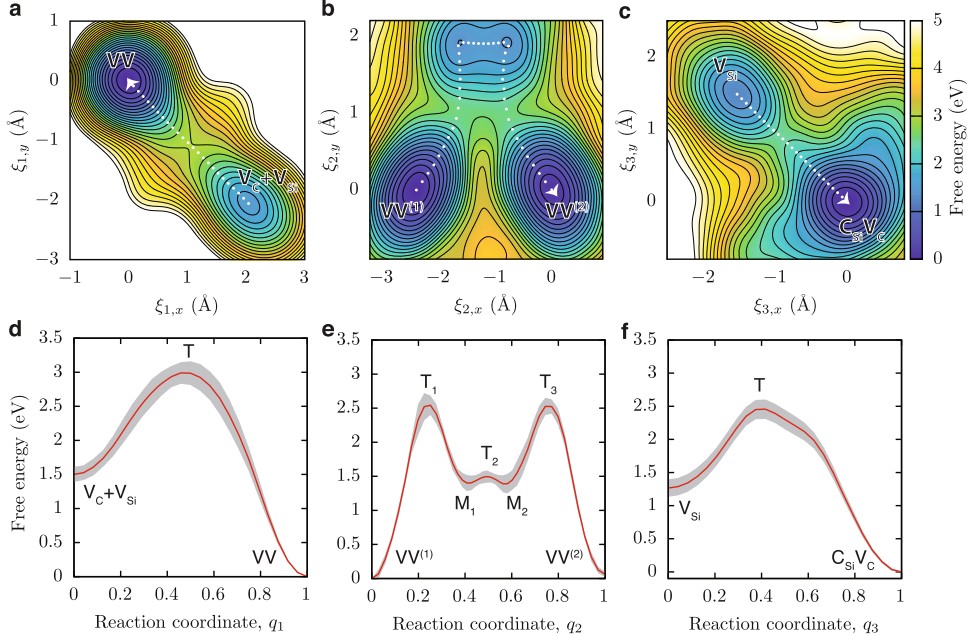

**Fig. 3 Free-energy landscapes of vacancy conversion processes for VV and $V_{Si}$.** Potentials of mean force (PMFs) were computed using enhanced sampling simulations with FPMD. Two-dimensional order parameters, $\xi_i = (\xi_{i,x}, \xi_{i,y})$, are used to describe the position of a carbon atom migrating toward a vacancy site, which is the primary mechanism for VV formation (**a, d**), VV reorientation (**b, e**), and $V_{Si}$ to $C_{Si}V_C$ conversion processes (**c, f**). **a–c** 2D-PMFs show free energy surfaces of vacancy-conversion processes at 1500 K, whose minimum free-energy pathways are marked by white dotted lines. **d–f** Free-energy profiles reveal intermediate (state $M_i$) and transition states (state $T_i$) along the reaction coordinates. The gray shaded regions denote the error in the PMF determined by block averaging. All three mechanisms are thermally activated processes with relatively high-energy barriers (>1.3 eV).

Fig. 6b) by ~0.4 eV. The PMF for the former process features two energetically equivalent local minima ($M_1$ and $M_2$) along the free-energy profile (Fig. 3d). By comparing the free-energy profiles for VV formation and reorientation processes (Fig. 3d, e), we find that the barrier for VV dissociation (VV → $V_{Si}$ + $V_C$) is ~0.5 eV higher than that required to change its orientation. This result suggests that the reorientation of divacancies can occur at a temperature lower than the temperature at which divacancies dissociate.

First-principles free-energy calculations reveal that the height of the barrier for $V_{Si}$ conversion into $C_{Si}V_C$ (~1.3 eV) is similar to that of the barrier for VV formation (~1.5 eV) (see Fig. 3d, f). Moreover, the activation energy for the formation of $C_{Si}V_C$ from $V_{Si}$ is lower than that of the migration of $V_{Si}$ (~3.0 eV barrier; see Supplementary Fig. 5b). Thus, $C_{Si}V_C$ formation is more likely to occur than silicon migration. Interestingly, the free energy of $C_{Si}V_C$ is lower than that of $V_{Si}$ by ~1.3 eV at 1500 K.

**Optimal temperatures for divacancy stability in as-grown and irradiated SiC.** To understand the behavior of divacancies at elevated temperatures, we first analyze the effect of annealing temperatures on the population of VVs in our classical MD simulations. As shown in Fig. 4a, the VV population decreases with increasing temperature above 1700 K, which is consistent with annealing experiments that measured the population of VV of as-grown SiC samples using electron paramagnetic resonance spectroscopy (EPR)[48]. We find that the kinetics of VV decay is faster in the presence of a higher-vacancy concentration in the sample (see Supplementary Fig. 8). Hence, it is likely that the VV decay observed experimentally is slower than the one found here, since the VV concentrations of the experimental samples are much lower than those considered in our simulations (see Methods). By fitting an Arrhenius equation to the experimental data, i.e., $y \propto \exp(-\Delta E/k_B T)$ (see Fig. 4a), we estimate the energy difference between a bound and a dissociated VV state to

be ~2.1 eV. This value is on a similar order of magnitude as the free-energy difference between VV and $V_{Si}$ + $V_C$ obtained from FPMD (~1.5 eV).

After collecting classical MD trajectories for 10 ns starting with stable VVs, we find that multiple vacancy defects are formed as divacancies dissociate (Fig. 4b), leading to the presence of $V_C$, $V_{Si}$, and $C_{Si}V_C$. The dominant species detected at high temperatures are carbon vacancies. Other defects ($V_{Si}$ and $C_{Si}V_C$) are present in smaller fractions (less than ~10% after 10 ns), as they can convert to $V_C$ via the following process: $V_{Si} \rightarrow C_{Si}V_C \rightarrow C_{Si} + V_C$. In order to validate our classical MD results, we calculated energy differences using DFT, and then we computed defect populations via kinetic Monte Carlo (KMC) simulations (see Supplementary Note 2). Similar to classical MD simulations, our KMC model shows that, at high temperatures, carbon vacancies are the dominant species (Fig. 4b). However, the KMC model predicts that the temperature required to dissociate $C_{Si}V_C$ is higher than that found in classical MD, because the $V_C$ migration barrier is higher in FPMD (~3.9 eV) than in classical MD (~2.5 eV). The results from MD and KMC simulations of double vacancies and the computed free-energy landscape of VV formation demonstrate that temperature facilitates both the creation and dissociation of VV. We expect the optimal temperature regime to anneal defects and generate VV to be between ~1000 K and ~1500 K or 1250 ± 250 K. This finding is consistent with recent annealing experiments carried out for irradiated or ion-implanted samples of 4H-SiC[6,44], in which PL data of VV also suggest that the concentration of VV defects is maximized at ~1200 K (Fig. 4c). More specifically, by fitting a Gaussian process model to experimental data, we estimate the optimal annealing temperature in experimentally irradiated samples to be ~1193 K.

**Compositional dependence in divacancy formation.** Our simulations and free-energy calculations show that the kinetics of the VV formation is controlled by the diffusion of

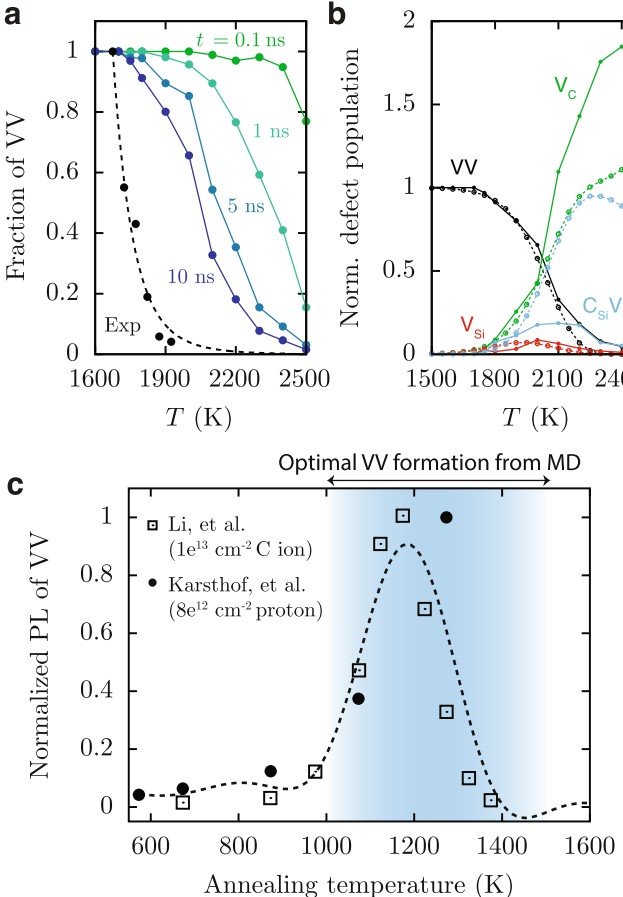

**Fig. 4 Effects of temperature on the divacancy formation and destabilization. a** Fraction of divacancies computed from classical MD simulations up to 10 ns, starting with only VVs. The black circles are experimental data, showing normalized EPR intensity of VV measured after annealing a nonirradiated 4H-SiC sample for 30 min[48]. The dotted black line is a fit to experimental data (see descriptions in the text). Both the simulations and the experiment indicate that divacancies begin to dissociate at ~1700 K. **b** Population of defects after VV dissociation. Solid and dotted lines indicate classical MD (10-ns long) and kinetic Monte Carlo (KMC) ($10^{15}k_0^{-1}$-long) simulations, respectively. Both sets of data demonstrate that $V_C$'s are the dominant defect species at high temperatures. At each temperature, the number of defects created from VV-dissociation process ($V_C$, $V_{Si}$, and $C_{Si}V_C$) is normalized by the initial number of VVs. **c** Photoluminescence (PL) intensity of VV versus annealing temperature based on data from recent experimental studies using irradiated or ion-implanted samples of 4H-SiC[6,44]. The PL intensity in each data set is normalized by the maximum signal measured. The type of implantation ion and the dose are shown in the legend. The dotted line is the fitted curve using a Gaussian process model, which predicts an optimum annealing temperature of ~1193 K. The temperature regime for optimal VV formation from MD simulations is between ~1000 K and ~1500 K (blue shaded region).

monovacancies, while the maximum number of divacancies is limited by the stability of $V_{Si}$. These results suggest that the annealing temperature and relative concentration of $V_{Si}$ to $V_C$ affect the mono-to-divacancy conversion efficiency. Experimentally, the composition of point defects in the sample can be changed by growing either carbon-rich or silicon-rich samples[49] or by controlling the Fermi level[21]. Thus, it is of interest to compute the mono-to-divacancy conversion efficiency at varying ratios of $V_{Si}$ to $V_C$ (see Methods and Supplementary Note 2).

In the KMC simulations carried out in the temperature range where the conversion $C_{Si}V_C \rightarrow C_{Si} + V_C$ occurs, we find that higher concentrations of $V_{Si}$ lead to an increase of the probability of VV formation, even in the absence of $V_C$ at $t = 0$. These results are in agreement with those of classical MD simulations at 1500 K (see Fig. 5a) and are intriguing, as one would expect no stable VV when the system has predominantly one type of vacancy ($V_C$ or $V_{Si}$). The results obtained at lower temperature are more intuitive as the KMC simulations show that the maximum number of divacancies is formed when there is an equal concentration of $V_{Si}$ and $V_C$ in the sample. These findings reveal that at high temperatures, the optimal condition to form VV would be to start with a large initial fraction of $V_{Si}$ to mobilize monovacancy migration and $C_{Si}V_C$ dissociation, so as to convert some $V_{Si}$ into $V_C$ and then generate additional $V_C$, as shown schematically in Fig. 5b. A low initial concentration of $V_{Si}$ reduces the VV-conversion efficiency (see Fig. 5c) because there are no energetically favored pathways to form $V_{Si}$ from $V_C$.

**MD simulations identify potential spin defects**. The generation of $V_{Si}$ and $C_{Si}V_C$ species during MD simulations of VV dissociation, validated by DFT-based KMC simulations (Fig. 4b), pointed out species that, in appropriate charged states, are interesting spin defects[10,14]. Prior studies have detected and characterized spin defects based on impurities and antisite defects using EPR measurements and DFT calculations at $T = 0$ K[50,51]. Remarkably, we also identified potential spin defects from MD simulations of divacancies at 2000 K and above through the process:   $VV \rightarrow V_C C_{Si} V_C \rightarrow C_{Si}V_C + V_C$   (see Supplementary Movie 5). Importantly, simulations show that these defects can be stabilized to room temperature (see Supplementary Note 3). Multiple structures of this vacancy complex are possible, depending on the relative orientation between $C_{Si}V_C$ and $V_C$, as well as the physical proximity between the two defects.

The results of hybrid DFT calculations[52] of the electronic structure of the neutral antisite-vacancy complexes shown in Fig. 6, indicate that $V_C C_{Si} V_C$ (Fig. 6b) and one of the structures for $[C_{Si}V_C + V_C]$, $[C_{Si}V_C + V_C]_{n=3}$ (Fig. 6c), consist of defect levels with localized electron densities. The lowest electronic configuration of these defect levels is a triplet, as the NV⁻ in diamond and VV⁰ in SiC. However, we note that in the antisite-vacancy complexes, the two carbon vacancies are separated by a carbon antisite and, therefore, their ground-state spin densities are more spatially delocalized than those of the VV state.

In the case of the $V_C C_{Si} V_C$ state, we find unoccupied defect levels within the bandgap close to the conduction band, similar to neutral VV spin defect in 3C-SiC[53] (see also Fig. 6a). These results suggest that $[C_{Si}V_C + V_C]_{n=3}$ and, especially, $V_C C_{Si} V_C$ are possible spin defects. Since $V_C C_{Si} V_C$ and $[C_{Si}V_C + V_C]$ are created when VV dissociates ($T > \sim 1700$ K), it is unlikely that these spin defects are simultaneously present with VVs in post-annealed samples. However, if present, high concentrations of these spin defects and/or isolated $C_{Si}$ species with localized electronic spins[54,55] may unfavorably decrease the VV coherence times[56].

Notably, our simulations did not incorporate any prior information on the formation mechanism of the spin defects found here. This highlights the potential of an atomistic approach to identify not only novel spin defects when combined with electronic-structure calculations but also microscopic mechanisms of their formation.

## Discussion

Our simulations of the relative stability of monovacancies and divacancies in 3C-SiC and their dynamics have revealed that, to generate a high concentration of VV in the material, a large ratio

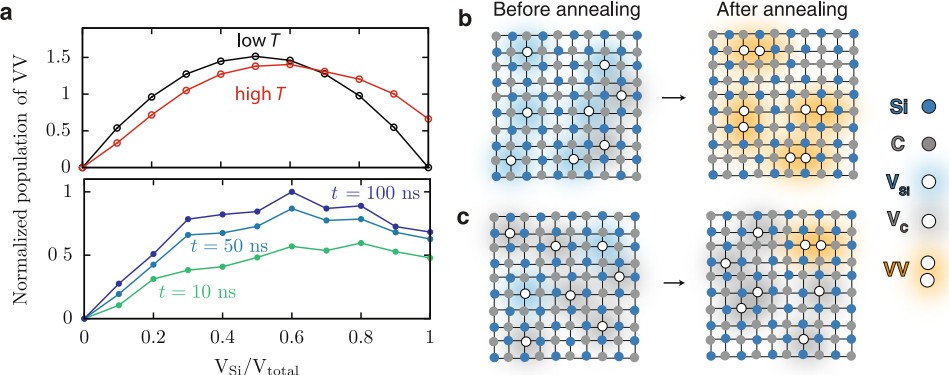

**Fig. 5 Dependence of VV-conversion efficiency on the initial composition of monovacancies. a** (top panel) KMC simulations showing changes in the long-time limit of VV population starting with varying concentrations of $V_{Si}$, i.e., the number of $V_{Si}$ over the total number of vacancies, at two different temperatures $T$ (black line at $T_{KMC} = 1500$ K, red line at $T_{KMC} = 2000$ K). (bottom panel) Classical MD simulations showing the time evolution of mono- to divacancy conversion up to 100 ns at $T_{MD} = 1500$ K, starting with different fractions of $V_{Si}$, i.e., the number of $V_{Si}$ over the total number of $V_{Si}$ and $V_C$. The VV formation favors higher concentrations of $V_{Si}$, agreeing with the high-temperature behavior of the KMC model. **b-c** Schematics for controlling divacancy formation by tuning the $V_{Si}$-to-$V_C$ ratio during thermal annealing at high temperature based on the results from panel **a**. **b** Samples having a greater number of $V_{Si}$ than $V_C$ prior to annealing (left) producing more VV after annealing (right). **c** Samples having a fewer number of $V_{Si}$ than $V_C$ prior to annealing (left), producing fewer VV after annealing (right).

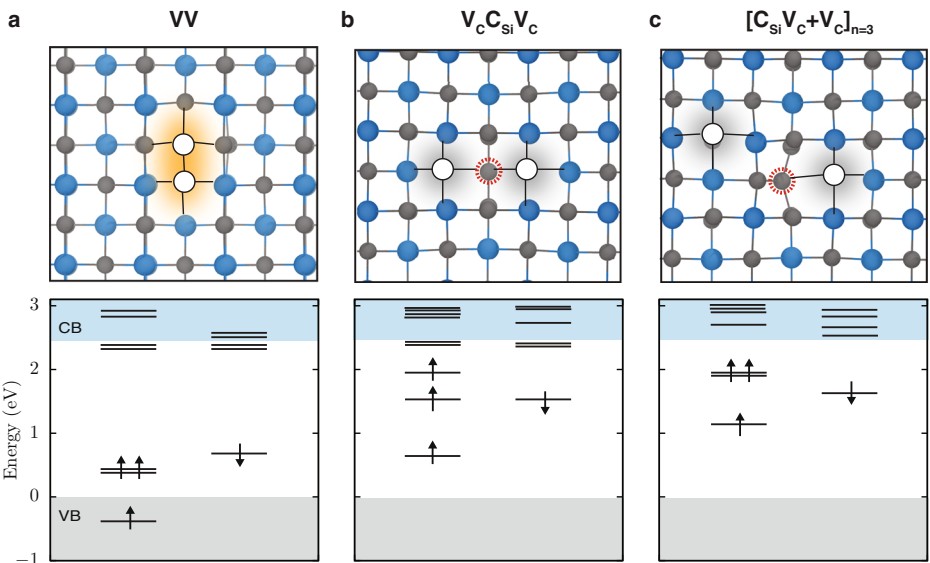

**Fig. 6 Electronic structures of $VV^0$ and antisite-vacancy complexes identified from MD simulations of VV dissociation.** Defect structures (top) and defect energy-level diagrams (bottom) are shown for neutral **a**, VV, **b**, $V_C C_{Si} V_C$, and **c**, $[C_{Si} V_C + V_C]_{n=3}$. For the $[C_{Si} V_C + V_C]$ defect shown here, the two carbon vacancies are separated by at least $n = 3$ atoms. For each defect, the lowest electronic configuration is a triplet state. Shaded gray areas indicate energy levels below the valence band (VB) and above the conduction band (CB). The spin-majority (spin-minority) channel is denoted by upward- (downward-) pointing arrows.

of silicon to carbon-vacancy concentration is needed. Several of the defect-transformation mechanisms observed in our simulations are consistent with and help explain annealing experiments, both in as-grown samples and irradiated/ion-implanted samples.

Divacancies are found to undergo crystallographic reorientation without dissociating. These findings indicate that the optimal annealing temperature for reorientation, whose control is of interest for sensing applications, is lower than that required to generate the defects.

Beyond having delineated the formation mechanisms of VV, $V_{Si}$, and $C_{Si} V_C$ spin defects, which arise from the dissociation of VV into $C_{Si} V_C$ and another $V_C$, we have also discovered plausible new spin defects and characterized their electronic structure using hybrid density-functional theory. Such defects provide new opportunities for quantum technology applications.

## Methods

**Classical MD simulations for divacancy dynamics.** Bulk 3C-SiC was modeled as a cubic supercell containing 4096 atoms with 3D periodic boundary conditions. Energy minimization and MD simulations were carried out using LAMMPS[57] and an environment-dependent interatomic potential (EDIP) force field[39], chosen after benchmarking several empirical potentials for SiC (Supplementary Fig. 1). We equilibrated a 3C-SiC structure at the target temperature $T$ and pressure of 1 atm in the NPT ensemble for 1 ns with a 1.0-fs time step. The system was further equilibrated for an additional 1 ns in the NVT ensemble. A Nose–Hoover thermostat and barostat were used to control temperature and pressure, respectively.

After the 2-ns long equilibration phase of the bulk crystal, three types of NVT-production runs were conducted with supercells containing one of the following: (1) a pair of $V_C$ and $V_{Si}$ and (2) a divacancy, and (3) 40 monovacancies with varying composition of monovacancies. In the first case, 20 independent 10-ns-long trajectories were sampled at each $T$ between 500 K and 2500 K, every 500 K. In the second case, 100 independent 10-ns-long trajectories starting with a single divacancy were sampled at each temperature between 1500 K and 2500 K, every 100 K. In the last case, a high-vacancy concentration (40/4096 = 1%) was used to increase the

probability of forming VVs from monovacancies within the simulation time of 100 ns. At each composition (ratio of $V_{Si}$ to $V_C$), 25 independent 100-ns-long trajectories were collected at 1500 K. The VV-conversion efficiency at time $t$ was calculated by dividing the number of divacancies at time $t$ by half of the number of monovacancies at $t = 0$. Overall, we performed over 2200 MD simulations of 10–100 ns each.

**Void-volume analysis for vacancy defects.** To identify and track the position of vacancy defects, we constructed Voronoi cells around each atom in a bulk crystal using VORO++[58]. The location of the vacancy defect was defined by the centroid position of an empty Voronoi cell after aligning the atoms in the bulk crystal to those in the MD snapshot. We computed the 3D void volume at the vacancy-defect site by calculating the distribution of voids on a discretized 3D grid with a spacing of $0.15 Å \times 0.15 Å \times 0.15 Å$. The excluded volume of each atom was approximated as a sphere with a radius $R$ (1.6 Å for carbon and 2.0 Å for silicon), where the ratio of the C and Si radii was the same as the ratio of their respective van der Waals radii.

**Determination of defect-energy levels using DFT calculations.** Electronic-structure calculations were carried out using DFT as implemented in the Quantum Espresso[59] and Qbox code[60,61]. Defects were modeled using $4 \times 4 \times 4$ or 512-atom supercells of 3C-SiC. The interaction between core and valence electrons was described using Optimized Norm-Conserving Vanderbilt pseudopotentials from the SG15 library[62] and we used a plane-wave basis with a kinetic energy cutoff of 55 Ry; the Brillouin zone was sampled with the Γ-point. For each defect calculation, a smearing with a width of 0.001 Ry was used in ground-state electronic-structure calculation, using the Marzari–Vanderbilt procedure[63]; geometries were optimized at the generalized gradient approximation (GGA) level of DFT using the Perdew–Burke–Ernzerhof (PBE) functional[64] in Quantum Espresso. The defect-level energy diagrams were obtained using the dielectric-dependent hybrid (DDH) functional with a self-consistent Hartree–Fock mixing parameter of $\alpha = 0.15$ for SiC[52] at the ground-state triplet configuration using the recursive bisection method[65] as implemented in the Qbox code.

**Free-energy calculations by coupling FPMD simulations with enhanced sampling.** The free-energy landscape[66–69] for defect migration and formation was computed using the combined force-frequency method or CFF-FPMD[36] as implemented in the software package SSAGES[70] coupled to the FPMD software Qbox. Unbiased FPMD and enhanced sampling simulations were performed in the NVT ensemble using the Bussi−Donadio−Parrinello thermostat[71] with a time step of 0.967 fs at $T = 1500$ K. To evaluate energy and forces, DFT calculations were carried out using the PBE functional using plane waves with a 40-Ry kinetic energy cutoff and the Γ-point to sample the Brillouin zone. We used the same pseudo-potential as in the defect energy-level calculations. For computational efficiency, the system was modeled with a 216-atom cell (see Supplementary Note 4 for the finite-size effect). Defect dynamical properties were simulated for the ground-state triplet configuration. The total charge of the supercell was neutral; however, charge rearrangements (i.e., charges near individual vacancy sites) during vacancy migration are allowed to occur in FPMD simulations. The $x$- and $y$-coordinates of a mobile atom displacing a vacancy site were used as the collective variables (CV), e.g., $\xi_{i,x} = X_{C_i}$ and $\xi_{i,y} = Y_{C_i}$ for the $V_C$ migration. During CFF-FPMD simulations, running average of forces and frequencies was recorded on a 2D grid with a grid spacing of 0.21 Å along $\xi_{i,x}$ and $\xi_{i,y}$. Bayesian-regularized artificial neural networks (BRANN) with two hidden layers (8 and 6 nodes in the first and the second layers, respectively) were used. For each free-energy calculation, up to 63 walkers/replicas were employed, and simulations were carried out up to 30 ps per walker. To compute the mean free-energy path (MFEP), we fitted the 2D-PMF to a BRANN to enable a smooth interpolation of free energies from a discretized 2D-CV grid[36]. The nudge elastic-band method[72] with a spring constant of 5 eV/Å was applied to the BRANN-fitted PMF to find the MFEP between the initial and final states.

## Data availability
Data that support the findings of this study will be available through the Qresp[73] curator at https://paperstack.uchicago.edu/explorer.

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

## Acknowledgements

We thank Chris Anderson, Nazar Delegan, and He Ma for insightful comments and discussions, Joe Heremans for a careful reading of the paper and helpful discussions, and François Gygi for help in setting up FPMD calculations using the Qbox code. We gratefully acknowledge the use of Bebop in the Laboratory Computing Resource Center at Argonne National Laboratory; the computational resources at the University of Chicago Research Computing Center; and the Argonne Leadership Computing Facility, a DOE Office of Science User Facility supported under Contract DE-AC02-06CH11357, provided by the Innovative and Novel Computational Impact on Theory and Experiment (INCITE) program. This work was supported by the Midwest Integrated Center for Computational Materials (MICCoM) as part of the Computational Materials Science Program funded by the US Department of Energy, Office of Science, Basic Energy Sciences, and Materials Sciences and Engineering Division, through Argonne National Laboratory, under Contract No. DE-AC02-06CH11357. A.Y. gratefully acknowledges support from the National Institute of Allergy and Infectious Diseases of the NIH under grant F32 AI150208.

## Author contributions

E.M.Y.L., J.J.d.P. and G.G. designed research. E.M.Y.L. performed research. E.M.Y.L. and A.Y. contributed methods, simulation code, and analytic tools. E.M.Y.L. and A.Y. analyzed data. E.M.Y.L., A.Y., J.J.d.P. and G.G. wrote the paper.

## Competing interests

The authors declare no competing interests.
