## [Peer Review File · Nature Communications]

REVIEWER COMMENTS

Reviewer #1 (Remarks to the Author):

The authors employed both classical molecular dynamics simulations and density functional theory calculations to investigate spin defects in SiC. The former was used for the thermal stability of the defects, and the latter was used to identify the defect levels in the band gap. In my view, calculations were performed with care, and the manuscript was clearly written. I also expect that discussion in this manuscript will widen our understanding of spin defect formation in semiconductors. I would like to recommend this manuscript for publication in Nature Communications if the authors properly answer the following questions/comments.

(1) There is a discrepancy in the migration barrier between the classical MD and ab-initio calculation, while the authors attribute the thermal effect and the supercell size to the origin of the difference. I suggest performing a barrier calculation with classical potentials in a smaller cell to check whether the difference really comes from the thermal effect and/or the cell size.

(2) The authors suggest a new complex defect as a spin defect. Based on the classical MD calculation study, this defect will form along with the divacancy defects? If the calculation indicates the simultaneous formation of both defects, then its effect on the qubit formation for quantum technologies should be discussed.

Reviewer #2 (Remarks to the Author):

Qubits realized by defects in a solid is a hot topic these days, and any clues given by theory, to help achieving a controllable technology for their production, is most welcome. The present paper tries to provide this help by performing extended empirical force field MD simulations to study the formation and stability of divacancies (VV) in SiC. In addition, based on the outcome of the study, some new “qubit candidates” are suggested, and their properties are calculated using an advanced hybrid functional.

Although it is a bit of an overstatement (alas, not the only one in the paper) that “SiC is a widely used material in the semiconductor industry”, the choice of the system is well justified. Even if silicon carbide occupies only a niche market in the semiconductor industry, the know-how of

processing it in a well-controlled manner is available, which is not quite the case for the other favorite, diamond. Also, the VV defect in SiC has some advantages over the NV defect in diamond.

The choice of the subject is, therefore, definitely adequate for NC, and the claims made in the paper could justify publishing it. My problem is with the choice and validation of the method, which make those claims questionable to me.

Classical force fields are parameterized for some standard bonding coordination of the constituent elements. They are rarely used for defect studies, where non-standard coordination frequently occur – especially during diffusion. Therefore, applying any such method in this kind of study needs verification. In this particular case, the diffusion barrier of the single vacancies could have served as a test. Unfortunately, as the paper states:

“Although prior DFT studies of 3C- and 4H-SiC have predicted lower potential energy barriers for VSi migration than VC migration,^{26,29,31} free energy calculations based on the empirical force field of 3C-SiC in this work estimate that the VSi migration barrier is higher than that of VC.”

The authors note that:

“...the DFT calculations were carried out at $T = 0$ K, thus neglecting important temperature effects; furthermore, these studies employed supercells containing 216 to 576 atoms, much smaller than the size of the supercell used in our MD simulations (4096 atoms)”.

However, none of these statement explain the discrepancy. For one thing, there are DFT calculations considering the effect of temperature (DOI: 10.1103/PhysRevB.68.155208), which does not change the sequence $VC > VSi$ between the barriers (up to 1800K). Second, taking also into account the error of LDA/GGA, the difference between the two barriers, ~ 1 eV, is significant. There is no way for the relaxation energy between a 216- and a 4096-atom supercell to change that (see, e.g., Fig.1 in Ref.[31] of the MS).

I can only agree with the explanation given by the author's

“These discrepancies may be due to different treatments of interatomic interactions in DFT and the empirical force field.”

Unfortunately, unless proven otherwise, it is more likely that it is the empirical force field which fails, not the DFT. Of course, the paper quoted above, as well as Ref.[26,27] of the MS, are rather old, and since then, functionals got better. The authors should use their dielectric-dependent hybrid functional to calculate the barriers in a 216 and in a 512 atom cell (that is computationally still feasible). From those, one can estimate the trend and magnitude of the size effect. Since it is pretty sure that the temperature will not change the sequence, unless these calculations yield $V_{Si} > V_C$, the results of this MD study can be discarded.

The authors justify their not doing so by qualitative agreement to some experimental facts. In my opinion, that is not sufficient. In fact, it would have been nice if the paper emphasized the conclusions more which have not been known beforehand.

To me the main novelty part could be in the proposal for new spin defects, complexes involving self-interstitials, except that the authors ignore important papers of the existing literature on that (and also on VV), which makes it hard to establish the real novelty.

It should also be noted that the migration barriers strongly depend on the charge state of the two vacancies, while VC and VSi cannot be both stable in the neutral charge state at the same Fermi-level position. It is unclear to me, how an empirical forcefield MD can handle this.

In summary, unless the applied method is validated, and the question of the Fermi-level dependence is cleared, I cannot recommend publication of the results. Also a better synopsis of the state of the art is needed.

Reviewer #3 (Remarks to the Author):

This manuscript reports results which potentially meet the requirement for the publication in Nature Communication. Indeed, the previous investigation on this subject usually focuses on the properties of potentially interesting defect-type or impurities-type centre, whilst their generation mechanism and the related suggestions for their control in the hosting material is barely studied. Although the research is strongly based on the application of a semiempirical approach, which needs of further verification for the full validation, the scenario here reported seem reliable with some confirmation from the experimental findings. Moreover, atomic and electronic structure of interesting complexes emerging from the high temperature dynamics are partially investigated with ab-initio based

approaches, providing further information for additional studies of quantum control protocols based on spin only Hamiltonians of the centre coupled with nuclear baths of the 3C-SiC material (e.g. for the scope the hyperfine coupling tensor can be calculate for nuclear spin close to the complex within the same ab-initio scheme).

In spite to the cited merits and the overall soundness of the results, the research has some technical and formal aspects which are not completely elucidated. I will try to indicate the points which need obligatory clarifications, further calculations or a thorough analysis.

The study shows that C_{Si} anti-sites emerge are necessary configuration of the V dynamics due to the conversion of the barely mobile V_{Si} (with MD!). This is an important aspect for the quantum control applications of the VV center: Its configuration should be often accompanied by at least one CSi present in the neighbourhood (not in a complex configuration as the ones studied with DFT) which can also strongly modify the spin evolution. Could you discuss the implication of this issue?

The discrepancy between the diffusion barrier of V_C and V_{Si} calculated with DFT and the one estimated with this study is briefly discussed. Actually, some justifications are not convincing. The smaller size of the DFT supercell should not be the cause of the discrepancies since the size dependence accuracy of DFT can be evaluated and it is below the calculated energy barrier difference. Moreover, the claimed temperature effects should be somewhat quantified. Finally, I wonder which aspect of the global scenario are still valid assuming a similar diffusivity for V_C and V_{Si}.

In figure 4.a) the predicted and real annealing of the VV defect is compared but at different temperature and time scales. I believe that an approximated scaling of the simulated kinetic at lower temperature (e.g. with an Arrhenius plot) could be attempted to check if the τ_{sim} annealing time constant at high T is compatible with 30min one of the experiments. Moreover, I believe that the role of the initial V density in simulation and experiment should be discussed: simulation could accelerate the kinetics dealing with a strongly different V density with respect that generated with irradiation.

Polytype choice (3C) seems a way to avoid the multiplicity of V-V type configurations of the 4H or 6H ones, which in turn can currently synthesised with a high purity. This seems a weak motivation. Actually, the technology relevance of the 3C-SiC will probably increase in the future and the possibility to obtain high quality 3C material make the study of V-systems in the 3C-SiC material “per se” relevant (see e.g. the results of the CHALLENGE project at the web page h2020challenge.eu). Could you rephrase the introduction in this sense?

A 3x3x3 conventional 3C-SiC cubic super cell is used for the DFT calculation. I wonder if the supercell size is sufficient for the accurate determination of the energy level of the defect (especially for the CSiVC+V_c cases). Could you discuss this issue? Moreover, their values with respect to some reference points (e.d. E_V or E_C) of the ideal band structures. Probably, the results are qualitatively correct (e.g. the triplet kind of state) but the accuracy should be improved if wish to use ab-initio approach to calibrate parameter in spin models.

Melting point at page 4 is not a suitable definition for SiC polytypes which cannot have a stable phase coexistence with the liquid counterpart. I would suggest referring to decomposition temperature as in the fig S1 caption.

Response to Reviewers

We thank all reviewers for their thorough and constructive comments on the manuscript. Prompted by the editorial and reviewers' comments on validating classical MD simulations results with DFT, we performed further calculations and significantly revised the manuscript and the Supporting Information (SI).

Here, we reproduce the reviewers' comments in **blue**, our responses to these comments appear in **black**. Corresponding changes in the main text and the SI are highlighted in **red**.

Referee Comments:

Reviewer # 1:

The authors employed both classical molecular dynamics simulations and density functional theory calculations to investigate spin defects in SiC. The former was used for the thermal stability of the defects, and the latter was used to identify the defect levels in the band gap. In my view, calculations were performed with care, and the manuscript was clearly written. I also expect that discussion in this manuscript will widen our understanding of spin defect formation in semiconductors. I would like to recommend this manuscript for publication in Nature Communications if the authors properly answer the following questions/comments.

We thank the reviewer for a thorough and positive assessment of our work.

1. There is a discrepancy in the migration barrier between the classical MD and ab-initio calculation, while the authors attribute the thermal effect and the supercell size to the origin of the difference. I suggest performing a barrier calculation with classical potentials in a smaller cell to check whether the difference really comes from the thermal effect and/or the cell size.

We have added a new section to the SI (Supplementary Note 1) to discuss discrepancies in migration barriers between classical MD and First Principles MD (FPMD) based on DFT. We have also significantly revised our manuscript, notably the introduction (pp.3-4), the results section on free energy landscapes (pp. 6-7), and the results section on changes in defect populations (pp. 8-9).

To check the differences originating from thermal effects and cell size, we performed additional classical MD calculations. In particular, we compared free energies and potential energy barriers computed with smaller cell sizes (216-atom) with those obtained in our original system (4096-atom). The reported energies for V_{Si} and V_C migration barriers are summarized in Supplementary Table 1. We find that the size-dependence of the free energy for a single vacancy migration event is within ~ 0.15 eV, which is comparable to $k_B T$ at $T = 1500$ K (~ 0.12 eV). Estimated migration barriers at 0 K indicate that the thermal effect can be as large as ~ 0.3 eV, but smaller than the difference in the energy barrier between V_C and V_{Si} migration processes (~ 1.0 eV). Thus, we concluded that the thermal effects and cell sizes are insufficient to explain the discrepancy in the migration barrier between our classical MD simulations and prior DFT calculations at 0 K (e.g., by R. Defo, et al. *Phys. Rev. B.* 98, 104103, 2018; and M. Bockstedte, et al. *Phys. Rev. B.* 68, 205201, 2003).

To map the free energy landscapes of defect dynamics at the DFT level, we performed enhanced sampling simulations coupled with FPMD. We then compared computed free energies obtained from FPMD and classical MD (see Supplementary Table 2). We find quantitative differences between the results of the two methods, but the overall qualitative results are the same for classical and FPMD. Specifically, VV has a lower free energy than dissociated states (*i.e.*, those of the V_{Si} and V_C); $C_{Si}V_C$ has a lower free energy than V_{Si} ; and the V_{Si} -to- $C_{Si}V_C$ conversion process has a lower free energy barrier than the single vacancy (V_{Si} or V_C) migration processes. Also, the free energy barrier for VV reorientation is lower than that of VV dissociation into V_{Si} and V_C . Therefore, the key findings of our work—(1) VV formation is limited by V_{Si} stability and (2) VV reorientation occurs without dissociation—are now also confirmed by using FPMD.

However, we found discrepancies in the migration barriers of V_{Si} and V_C between classical MD and FPMD, as the latter predicts that the migration barrier of V_{Si} is lower than that of V_C . Therefore, to analyze changes in defect populations obtained at the DFT-level, relative to what originally reported at the classical level, we performed additional calculations using a kinetic Monte Carlo (KMC) method, which was built using the energetics from DFT results (see Supplementary Note 2). The KMC results (see Figures 4b and 5a in the main text) show a defect population analysis in qualitative agreement with that obtained from classical MD—VVs are converted to V_C , and high V_{Si} -to- V_C ratios favor the formation of VV formation. The primary, quantitative difference between classical MD and KMC results is that the latter predicts that a higher temperature is needed to dissociate $C_{Si}V_C$ than in classical MD, because the V_C migration barrier found using DFT (~ 3.9 eV) is higher than in classical MD (~ 2.5 eV).

2. The authors suggest a new complex defect as a spin defect. Based on the classical MD calculation study, this defect will form along with the divacancy defects? If the calculation indicates the simultaneous formation of both defects, then its effect on the qubit formation for quantum technologies should be discussed.

We thank the reviewer for this interesting question. In the classical MD simulations, new spin defect complexes are formed via dissociation of VVs. Therefore, it is unlikely that both VV and new spin defects will form simultaneously. However, if these species are present in high concentrations, they may decrease the spin coherence times of VV. We have now edited the last paragraph of the results section (p.11, lines 296-300) to discuss this point.

Reviewer #2:

Qubits realized by defects in a solid is a hot topic these days, and any clues given by theory, to help achieving a controllable technology for their production, is most welcome. The present paper tries to provide this help by performing extended empirical force field MD simulations to study the formation and stability of divacancies (VV) in SiC. In addition, based on the outcome of the study, some new “qubit candidates” are suggested, and their properties are calculated using an advanced hybrid functional.

Although it is a bit of an overstatement (alas, not the only one in the paper) that “SiC is a widely used material in the semiconductor industry”, the choice of the system is well justified. Even if silicon carbide occupies only a niche market in the semiconductor industry, the know-how of processing it in a well-controlled manner is available, which is not quite

the case for the other favorite, diamond. Also, the VV defect in SiC has some advantages over the NV defect in diamond.

The choice of the subject is, therefore, definitely adequate for NC, and the claims made in the paper could justify publishing it. My problem is with the choice and validation of the method, which make those claims questionable to me.

We thank the reviewer for a positive outlook on the choice of this topic. In our response to Reviewer 1's Question 1, we describe the additional calculations using first-principles molecular dynamics (FPMD) based on DFT, which are now added to the revised manuscript and compared with classical MD simulation results (see p.5, lines 129-132).

Classical force fields are parameterized for some standard bonding coordination of the constituent elements. They are rarely used for defect studies, where non-standard coordination frequently occur – especially during diffusion. Therefore, applying any such method in this kind of study needs verification. In this particular case, the diffusion barrier of the single vacancies could have served as a test. Unfortunately, as the paper states:

“Although prior DFT studies of 3C- and 4H-SiC have predicted lower potential energy barriers for VSi migration than VC migration,^{26,29,31} free energy calculations based on the empirical force field of 3C-SiC in this work estimate that the VSi migration barrier is higher than that of VC.”

The authors note that:

“...the DFT calculations were carried out at $T = 0$ K, thus neglecting important temperature effects; furthermore, these studies employed supercells containing 216 to 576 atoms, much smaller than the size of the supercell used in our MD simulations (4096 atoms)”.

However, none of these statement explain the discrepancy. For one thing, there are DFT calculations considering the effect of temperature (DOI: 10.1103/PhysRevB.68.155208), which does not change the sequence $VC > VSi$ between the barriers (up to 1800K). Second, taking also into account the error of LDA/GGA, the difference between the two barriers, ~ 1 eV, is significant. There is no way for the relaxation energy between a 216- and a 4096-atom supercell to change that (see, e.g., Fig.1 in Ref.[31] of the MS).

In the revised manuscript, we have performed size and thermal effect calculations using a classical MD approach. As seen in our response to Reviewer 1's Question 1, we find that finite size and thermal effects can be as large as ~ 0.3 eV, but insufficient to explain the difference between V_{Si} and V_C migration barriers (~ 1 eV).

I can only agree with the explanation given by the author's

“These discrepancies may be due to different treatments of interatomic interactions in DFT and the empirical force field.”

Unfortunately, unless proven otherwise, it is more likely that it is the empirical force field which fails, not the DFT. Of course, the paper quoted above, as well as Ref.[26,27] of the MS, are rather old, and since then, functionals got better. The authors should use their dielectric-dependent hybrid functional to calculate the barriers in a 216 and in a 512 atom cell (that is computationally still feasible). From those, one can estimate the trend and

magnitude of the size effect. Since it is pretty sure that the temperature will not change the sequence, unless these calculations yield $V_{Si} > V_C$, the results of this MD study can be discarded.

Following the reviewer's suggestions to compute migration barriers from DFT, we performed enhanced sampling simulations with FPMD. The reviewer recommended performing migration barrier calculations with dielectric-dependent hybrid functionals. Although ultimately desirable, the use of hybrid-functionals for enhanced sampling with FPMD simulations is computationally very expensive. Importantly, we note that total energy differences (as those computed for our sampling simulations) are expected to be much more accurate than single particle eigenvalues, when using semi-local functionals. Going beyond a GGA description and using hybrid functionals is of course essential when predicting properties based on Kohn-Sham eigenvalues (e.g., band gap), but not as critical when predicting potential energy surfaces. Therefore, to compute vacancy migration barriers which rely on total energy differences, we performed enhanced sampling simulations using a GGA functional (PBE). We also performed additional DFT calculations to show that the relative total energy between VV and $V_{Si} + V_C$ is converged within 0.1 eV using the 216-atom cell, compared to the 512-atom cell (Supplementary Table 4). Hence, we used a 216-atom cell in our FPMD simulations.

The results of our simulations are summarized in Supplementary Table 2 and discussed in the revised manuscript (see the results section on free energy landscapes in pp. 6-8) and in our answer to Reviewer 1's Question 1.

The authors justify their not doing so by qualitative agreement to some experimental facts. In my opinion, that is not sufficient. In fact, it would have been nice if the paper emphasized the conclusions more which have not been known beforehand.

We performed additional calculations by developing a kinetic Monte Carlo (KMC) model, which was parameterized based on our DFT results to provide validations to our classical MD simulations. The KMC model is discussed in the revised manuscript (pp. 8-10 in the main text; see also Supplementary Note 2). As shown in Figures 4b and 5a, the results of the KMC model are in qualitative agreement with those of classical MD simulations. In particular, using both methods we find the following results: at high temperature, V_{Si} 's are converted to V_C , and a high V_{Si} -to- V_C ratio favors the formation of VV. Therefore, our major findings from classical MD hold true also when using DFT calculations, despite the difference in vacancy migration barriers obtained with the two methods.

We have also revised the manuscript (Introduction in pp. 3-4) to highlight conclusions obtained with DFT, and we discuss in detail the divacancy reorientation mechanisms that are important for quantum sensing applications and a new route to optimize VV formation by having a higher V_{Si} to V_C ratio.

To me the main novelty part could be in the proposal for new spin defects, complexes involving self-interstitials, except that the authors ignore important papers of the existing literature on that (and also on VV), which makes it hard to establish the real novelty.

We thank the reviewer for noting the novel parts of our paper regarding the formation of new spin defects. We have revised the last paragraph of the results section (p. 11, lines 302-304) to clarify the novelty of our work in using dynamical simulations combined with

DFT to not only propose new spin defects but also probe their formation mechanisms, in contrast to prior computational studies based on DFT at $T = 0$ K.

We have also added the following references to the main text (p.10) as we introduce previous studies pertaining to the identification and characterization of new spin defects:

50. P. Carlsson, N. T. Son, A. Gali, J. Isoya, N. Morishita, T. Ohshima, B. Magnusson, and E. Janzén. “EPR and ab initio calculation study on the E14 center in 4H- and 6H-SiC” *Phys. Rev. B*, 82, 235203, 2010

51. K. Szasz, V. Ivady, I. A. Abrikosov, E. Janzen, M. Bockstedte, and A. Gali. “Spin and photophysics of carbon-antisite vacancy defect in 4H silicon carbide: A potential quantum bit” *Phys. Rev. B*, 91, 121201, 2015

It should also be noted that the migration barriers strongly depend on the charge state of the two vacancies, while VC and VSi cannot be both stable in the neutral charge state at the same Fermi-level position. It is unclear to me, how an empirical forcefield MD can handle this.

The reviewer correctly noted that classical MD simulations cannot describe changes in charge states of individual defects. To compute the energetics at finite temperature, we carried out enhanced sampling simulations coupled with FPMD at a value of the Fermi-level known to favor the formation of neutral divacancies. We note that during FPMD simulations, charge rearrangements are allowed; in particular, charges near individual vacancy sites may undergo rearrangement when vacancies migrate, as in the $V_{Si} + V_C \rightarrow VV$ process. We discuss the Fermi level and charge rearrangements allowed by FPMD on p.7 (lines 183-184) and p. 14 (lines 385-386) of the revised manuscript.

In summary, unless the applied method is validated, and the question of the Fermi-level dependence is cleared, I cannot recommend publication of the results.

As discussed in our responses to Reviewer 1’s Question 1, we have now shown that our key findings from classical MD results agree with FPMD simulations results. Moreover, we have considered changes to the local charge of individual defects using FPMD.

Also a better synopsis of the state of the art is needed.

To provide a better synopsis of the state of the art, we have now cited the latest literature on spin defects in SiC in the main text on pp. 2-3:

6. Li, Q. *et al.* Room temperature coherent manipulation of single-spin qubits in silicon carbide with a high readout contrast. *National Science Review* (2021) doi:10.1093/nsr/nwab122.

21. Son, N. T. & Ivanov, I. G. Charge state control of the silicon vacancy and divacancy in silicon carbide. *Journal of Applied Physics* **129**, 215702 (2021).

26. Pavunny, S. P. *et al.* Arrays of Si vacancies in 4H-SiC produced by focused Li ion beam implantation. *Scientific Reports* **11**, 3561 (2021).

Reviewer #3:

This manuscript report results which potentially meet the requirement for the publication in Nature Communication. Indeed, the previous investigation on this subject usually focuses on the properties of potentially interesting defect-type or impurities-type centre, whilst their generation mechanism and the related suggestions for their control in the hosting material is barely studied. Although the research is strongly based on the application of a semiempirical approach, which needs of further verification for the full validation, the scenario here reported seem reliable with some confirmation from the experimental findings. Moreover, atomic and electronic structure of interesting complexes emerging from the high temperature dynamics are partially investigated with ab-initio based approaches, providing further information for additional studies of quantum control protocols based on spin only Hamiltonians of the centre coupled with nuclear baths of the 3C-SiC material (e.g. for the scope the hyperfine coupling tensor can be calculate for nuclear spin close to the complex within the same ab-initio scheme).

In spite to the cited merits and the overall soundness of the results, the research has some technical and formal aspects which are not completely elucidated. I will try to indicate the points which need obligatory clarifications, further calculations or a thorough analysis.

We thank the reviewer for the positive and constructive evaluation, especially on the implications of the spin defects identified here to quantum control protocols.

1. The study shows that C_{Si} anti-sites emerge are necessary configuration of the V dynamics due to the conversion of the barely mobile V_{Si} (with MD!). This is an important aspect for the quantum control applications of the VV center: Its configuration should be often accompanied by at least one C_{Si} present in the neighbourhood (not in a complex configuration as the ones studied with DFT) which can also strongly modify the spin evolution. Could you discuss the implication of this issue?

This is an interesting point that requires detailed investigations beyond the scope of the present work, that we intend to carry out in the future using recently developed methods to compute coherence times (M. Onizkuk, G. Galli, *et al. PRX Quantum* 2, 010311, 2021). It is reasonable to expect that the presence of C_{Si} in high concentration may decrease VV spin coherence times. However, further studies are necessary to address this issue that is now only mentioned briefly on p.11 (lines 298-300) of the revised manuscript.

2. The discrepancy between the diffusion barrier of V_C and V_{Si} calculated with DFT and the one estimated with this study is briefly discussed. Actually, some justifications are not convincing. The smaller size of the DFT supercell should not be the cause of the discrepancies since the size dependence accuracy of DFT can be evaluated and it is below the calculated energy barrier difference. Moreover, the claimed temperature effects should be somewhat quantified. Finally, I wonder which aspect of the global scenario are still valid assuming a similar diffusivity for V_C and V_{Si}.

We performed additional calculations to estimate both thermal and finite-size effects in classical MD simulations, and we find that these are insufficient to explain the differences in V_C and V_{Si} diffusion barriers found in classical MD and FPMD (please also see our response to Reviewer 1's Question 1). Hence, we carried out enhanced sampling simulations with FPMD to compute the free energies for defect migration, formation, and

conversion, and we compared the results to those of classical MD simulations. Notably, we find that V_{Si} has a higher free energy than $C_{Si}V_C$, and that the conversion of V_{Si} to $C_{Si}V_C$ has a lower free energy barrier than that of the V_{Si} migration. Therefore, both FPMD and classical MD find that V_{Si} is a kinetically trapped state. Hence, even if a similar diffusivity for V_C and V_{Si} is assumed, the formation of VV by association of monovacancies is hindered by the conversion of V_{Si} to $C_{Si}V_C$.

3. In figure 4.a) the predicted and real annealing of the VV defect is compared but at different temperature and time scales. I believe that an approximated scaling of the simulated kinetic at lower temperature (e.g. with an Arrhenius plot) could be attempted to check if the τ_{sim} annealing time constant at high T is compatible with 30min one of the experiments. Moreover, I believe that the role of the initial V density in simulation and experiment should be discussed: simulation could accelerate the kinetics dealing with a strongly different V density with respect that generated with irradiation.

We fitted the fraction of VV versus temperature, T , to an Arrhenius equation of the form, $y \propto \exp(-\Delta E/k_B T)$, where $\Delta E = E(VV \text{ dissociated}) - E(VV \text{ bound})$, as shown in Fig. 4a. This allowed us to estimate ΔE to be ~ 2.1 eV, which is within of the same order of magnitude as the free energy difference between VV and $V_{Si} + V_C$ obtained from FPMD (~ 1.5 eV). We now discuss this finding in the main text of the revised manuscript (p. 8, lines 225-228).

We agree that different vacancy densities in classical MD simulations (two V's in a 4096-atom supercell or 0.04 %) and in experiments (on the order of parts-per-million in concentrations or $\sim 1e-4$ % at most) may lead to different kinetics in the VV decay. To address this issue, we used kinetic Monte Carlo (KMC) simulations based on energetics obtained from DFT calculations (see Supplementary Note 2) and simulated the VV decay at two different concentrations: $1e-3$ and $1e-4$ % VV present at time $t=0$ (see Supplementary Fig. 8). We find that at higher vacancy concentrations, VV population decays faster. We now discuss this point in the main text (p. 8, lines 221-225) and in the SI (Supplementary Fig. 8) of the revised manuscript.

4. Polytype choice (3C) seems a way to avoid the multiplicity of V-V type configurations of the 4H or 6H ones, which in turn can currently synthesized with a high purity. This seems a weak motivation. Actually, the technology relevance of the 3C-SiC will probably increase in the future and the possibility to obtain high quality 3C material make the study of V-systems in the 3C-SiC material "per se" relevant (see e.g. the results of the CHALLENGE project at the web page h2020challenge.eu). Could you rephrase the introduction in this sense?

We thank the reviewer for the suggestion. We have revised our introduction on p.4 (lines 106-108) to better motivate the choice of 3C-SiC and cite additional references related to work described in the Challenge Project on 3C-SiC.

37. Agati, M. et al. Growth of thick [1 1 1]-oriented 3C-SiC films on T-shaped Si micropillars. Mater. Des. 208, 109833 (2021).

38. Fisticaro, G. et al. Genesis and evolution of extended defects: The role of evolving interface instabilities in cubic SiC. Appl. Phys. Rev. 7, 021402 (2020).

5. A 3x3x3 conventional 3C-SiC cubic super cell is used for the DFT calculation. I wonder if the supercell size is sufficient for the accurate determination of the energy level of the defect (especially for the CSiVC+Vc cases). Could you discuss this issue? Moreover, their values with respect to some reference points (e.d. E_V or E_C) of the ideal band structures. Probably, the results are qualitatively correct (e.g. the triplet kind of state) but the accuracy should be improved if wish to use ab-initio approach to calibrate parameter in spin models.

We repeated our DFT calculations using a larger, 512-atom supercell as shown in Figure 6, and edited our methods section in the main text (p. 13) to describe these new calculations. It appears that our results from 216-atom (Supplementary Fig. 4) qualitatively reproduce the 512-atom results, as the reviewer noted.

6. Melting point at page 4 is not a suitable definition for SiC polytypes which cannot have a stable phase coexistence with the liquid counterpart. I would suggest referring to decomposition temperature as in the fig S1 caption.

We have changed melting point to decomposition temperature in Supplementary Fig. 1 caption.

REVIEWERS' COMMENTS

Reviewer #1 (Remarks to the Author):

The authors revised the manuscript significantly according to the comments, so I would like to recommend the manuscript for publication in Nature Communications.

Reviewer #3 (Remarks to the Author):

The authors have properly addressed all my previous criticisms: Both responses and revision are appropriate and exhaustive. I believe that the revised version can be published in the present form in NC.

Response to Reviewers

Reviewer #1:

The authors revised the manuscript significantly according to the comments, so I would like to recommend the manuscript for publication in Nature Communications.

Reviewer #3:

The authors have properly addressed all my previous criticisms: Both responses and revision are appropriate and exhaustive. I believe that the revised version can be published in the present form in NC.

We thank all reviewers for the positive and supportive comments.